# Specific Neural Mechanisms Underlying Humans’ Processing of Information Related to Companion Animals: A Comparison with Domestic Animals and Objects

**DOI:** 10.3390/ani15213162

**Published:** 2025-10-31

**Authors:** Heng Liu, Xinqi Zhou, Jingyuan Lin, Wuji Lin

**Affiliations:** Institute of Brain and Psychological Sciences, Sichuan Normal University, No. 5, Jing’an Road, Jinjiang District, Chengdu 610068, Chinazhxq@sicnu.edu.cn (X.Z.); linjingyuan921@126.com (J.L.)

**Keywords:** companion animals, attachment, precuneus, functional connectivity, human–animal bond, emotion

## Abstract

**Simple Summary:**

Humans exhibit special brain reactions when processing information about companion animals (like cats or dogs), but we do not fully understand how this works. To figure it out, we studied 40 people using functional magnetic resonance imaging—a tool that hows brain activity—while they looked at four types of images (companion animals, other animals, pleasant objects, common objects) and judged each image’s category. We found three key things: first, looking at companion animals activated specific brain areas (linked to seeing, feelings, and attachment); second, the connections between these brain areas changed in a special way; third, pet owners and non-pet owners had clear differences in how some brain areas connect internally. This study confirms our brains react uniquely to companion animals. It helps us understand how companion animals affect our brain activity during interactions and can guide uses like animal-assisted therapy (where animals help with healing).

**Abstract:**

Humans show neural specificity in processing animal-related information, especially regarding companion animals. However, the underlying cognitive mechanisms remain poorly understood. This study’s main objective is to investigate human neural specificity in processing companion animal-related information, compared to other animal types and inanimate objects. Forty participants viewed four image types (companion animals, neutral animals, positive objects, neutral objects) during functional magnetic resonance imaging (fMRI) scans and judged image categories. T-test results showed: 1. Processing companion animal-related information elicited specific brain activation in the right Inferior Parietal Lobe (right IPL), right Middle Occipital Gyrus (right MOG), left Superior Frontal Gyrus (left SFG), and left Precuneus (left PCu) (<0.05). 2. Generalized Psychophysiological Interaction (gPPI) analysis revealed specific functional connectivity changes between relevant brain regions during companion animal info processing (<0.05). 3. Dynamic Causal Modelling (DCM) analysis showed significant intrinsic connectivity differences between pet owners and non-pet owners: specifically, left IPL to left PCu and right ACC to right MOG (posterior probability, Pp > 0.95). The results of this study demonstrate that humans exhibit distinct neural specificity when processing information related to companion animals compared with livestock and inanimate objects. This neural specificity involves brain regions linked to higher-order cognitive functions (e.g., visual processing, emotion, and attachment), all of which are integral components of the human attachment network. These regions are part of the human attachment network, and their functional role likely relates to attachment mechanisms. These findings help clarify companion animals’ impact on human neural activity during human–animal interactions and guide applications like animal-assisted therapy.

## 1. Introduction

“Why do humans get so attached to dogs and cats?” is one of the most challenging questions [1]. Studies have found that companion animals can reduce human loneliness and feelings of isolation, thereby affecting physical and mental health [2]. The presence of companion animals can motivate people to enjoy an active and healthy lifestyle [3], improve mental health [4,5], enhance quality of life [6,7]—for instance, pet therapy improves the sense of well-being of elderly residents in nursing homes and reduces their anxiety, depression, apathy, and loneliness [8] —improve physical fitness [9,10], and boost cognitive function [11,12]. Therefore, it is important to investigate how companion animals influence humans, particularly by revealing the underlying neural mechanisms.

Companion animals are widely recognized as a significant source of emotional support, and their presence can greatly enhance people’s mood [13]. Numerous studies have demonstrated that companion animals contribute positively to reducing stress [14,15,16], alleviating anxiety [17,18,19], mitigating depression [20,21], and enhancing life satisfaction [22]. In addition to improving mood, companion animals also induce specific neural activity. Companion animals activate reward-related brain regions such as the frontal lobe [23,24,25] and the amygdala [26]. They also enhance cardiovascular health by improving heart rate and blood pressure [27,28,29], regulate levels of oxytocin [30,31] and cortisol [32,33], and influence the secretion of other mood-related hormones, including serotonin, dopamine, phenylethylamine, endorphins, and prolactin [34,35,36,37].

The academic community has proposed multiple theories to explain the mechanisms underlying human–companion animal interactions. Attachment theory, originally proposed by John Bowlby, was initially developed to explain the intimate emotional bond between infants and their primary caregivers during early human life [38,39]. In recent years, this theory has been extended to account for the affiliative relationships between humans and their companion animals [40]. The Biophilia Hypothesis posits that humans possess an innate predisposition to connect with other living organisms—including animals—through evolutionary processes [41,42]. Social Support Theory emphasizes that companion animals function as key sources of “non-human social support” [40].

Findings from fMRI and functional Near-Infrared Spectroscopy (fNIRS) studies indicate that human neural activity involved in processing animal-related information is specific, engaging regions such as the frontal lobe [23,24,43], insula [44], and amygdala [26,45], and is independent of emotional valence and arousal [45]. Thus, through evolution and cohabitation with companion animals, humans may have developed specialized cognitive neural mechanisms, which enable companion animals to significantly influence human emotional well-being. This effect may arise from the attachment humans form towards companion animals [2].

Companion animals often play an important role as attachment figures in human life [46], such as dogs and cats [47]. Attachment theory was initially proposed by psychologist John Bowlby to explain how the emotional bond between infants and caregivers influences individual development from infancy to adulthood [38,48], highlighting the importance of the attachment figure in providing security, comfort, and protection. Extending to human–companion animal relationships, companion animals are also often viewed as a secure base, with individuals seeking emotional comfort through interactions with them [40]. When interacting with companion animals, activities in brain regions related to reward, emotion, and attachment (e.g., amygdala, frontal lobe) are enhanced [23,45], and adaptive regulation of related hormones (e.g., cortisol, oxytocin) occurs [31,49,50]. This indicates that companion animals, as attachment figures, can facilitate emotional regulation, enhance social functioning, and improve psychological adaptation [40,51]. An fMRI study compared brain activity patterns in mothers viewing pictures of their own children and dogs versus unfamiliar children and dogs, finding similar activation patterns in regions involved in emotion, reward, attachment, visual processing, and social cognition when mothers viewed their own child and dog images [26].

It can be seen that companion animals exhibit certain specificity in regulating human emotions, but the underlying mechanisms of this specificity are not yet well understood. Previous studies investigating the specific neural activity involved in processing animal-related information have often used neutral and negative animal images as experimental stimuli [52,53,54], with companion animals seldom being used. In studies exploring the neural activity triggered by companion animals, natural or other neutral stimuli are mostly used as the baseline [23,53,55]. As a result, the findings may include brain regions that are specifically activated by companion animal information, as well as those activated by positive and animal-related information, resulting in confounding factors. Therefore, when investigating the specific neural activity elicited by the processing of companion animal information in the brain, experimental designs should be used to eliminate interference from other confounding factors. In this study, we subtracted the four types of stimuli (see Figure 1) to eliminate confounding factors as much as possible and to explore the specific neural activity elicited by companion animals.

Therefore, we put forward the following hypothesis: First, companion animal-related information can induce specific neural activity in the human brain, a process involving brain regions related to attention, emotion, and attachment. (Hypothesis 1, Core Hypothesis). Secondly, people’s attitudes towards animals may lead to emotional changes regarding companion animals [56]. Thus, attitudes towards companion animals may modulate the above-mentioned specific neural activity (Hypothesis 2, Extended Exploration). Thirdly, “the key to various cognitive functions lies not in individual brain regions, but in the communication between them, cognitive functions involve interactions within dynamic neural networks” [57,58]. This view has gained increasing support among scholars [59,60,61]. Therefore, there is a necessity to explore the correlation between brain networks and human cognitive functions [62]. If Hypothesis 1 holds true, we will conduct exploratory functional connectivity analyses, further hypothesizing that companion animal information may elicit specific coordinated activity across multiple brain regions (Hypothesis 3, Extended Exploration).

## 2. Methods

The present study adopted a cross-sectional observational design, aiming to explore the neural differences in humans when processing different information stimuli. It is a prospective data collection study, where all data were systematically collected under a unified framework, and retrospective data were not used. In terms of grouping and randomization, it used a single-group design, with no grouping or intervention conducted, no randomization applied, and all participants received the same stimulus tasks and data collection. For blinding, a single-blinded design was adopted: neither participants nor data collectors were aware of the core experimental hypothesis. (Preregistration information. The relevant details of this study have been preregistered on Open Science Framework https://osf.io/yadxn/?view_only=597513d99a8045888d0439efbcb7c156, accessed on 21 November 2023)

### 2.1. Participants

Referring to the sample size used in previous similar studies [44] and the sample size estimation for fMRI neuroimaging research by Desmond and Glover (2002) [63], this study adopted convenience sampling and recruited 42 healthy right-handed college students (the detailed methodology for sample size calculation is provided in the Appendix A). After excluding 2 participants with excessive head motion, 40 participants remained for data analysis (self-reported gender: 35 females, 5 males). All participants had normal visual acuity or corrected-to-normal visual acuity; They had no history of major diseases, head trauma, or claustrophobia, and no history of any mental disorders, neurological diseases, cognitive impairments, or mood disorders; They had no history of major surgeries, and no implanted metals or medical devices in their bodies; And they had no fear response to images of cats, dogs, chickens and ducks [64]. Twenty-six participants reported currently (or previously) keeping pets. Among these, dogs were mentioned 16 times, cats 12 times, other mammals 4 times, reptiles 3 times, insects 2 times and fish 1 time. The experiment was approved by the ethics committee of the author’s institution and conducted in accordance with the ethical guidelines of the Declaration of Helsinki (Ethics Approval No.2023LS018). All participants signed informed consent forms (See Appendix A) and received 50 RMB for participation.

### 2.2. Experimental Materials, Scales and Procedure

#### 2.2.1. Preparation and Evaluation of Experimental Materials

The experimental materials were divided into four conditions: companion animals, neutral animals, positive objects and neutral objects. The companion animal condition included cats and dogs, the positive object condition included vases and sculptures, the neutral animal condition included chickens and ducks, and the neutral object condition included tables and chairs.

Rationale for Classifying the Four Stimulus Types Employed in This Study For animal stimuli: Cats and dogs are classified as companion animals, as they align with the established definition of “companion animals” and form intimate emotional bonds with humans (e.g., companionship, emotional comfort). By contrast, chickens and ducks are primarily perceived as “economic animals” (i.e., providers of meat and eggs) or “environmentally associated animals” (e.g., common poultry in rural areas). Their interactions with humans are dominated by functional contact and lack intimate attachment, thus classifying them as neutral animals. For object stimuli: Vases and sculptures are characterized by core attributes of decorativeness and aesthetic value—they symbolize an elegant lifestyle, elicit aesthetic pleasure, and carry positive cultural connotations—and are therefore classified as positive objects. Tables and chairs, as utilitarian daily tools, serve core functions of fulfilling practical needs (e.g., supporting writing activities, enabling seating). Their emotional association with humans is weak, so they are classified as neutral objects. (Additional details are provided in the online Appendix A).

Thirty participants were recruited to evaluate the valence of the materials to ensure that the valence of companion animals and positive objects was significantly higher than that of neutral animals and neutral objects, and that there was no significant difference in valence between companion animals and positive objects or between neutral animals and neutral objects. Some images were selected as practice trial materials, while the remaining images were used as formal experimental materials. The practice materials were not included in the formal experiment (see Appendix A for details).

#### 2.2.2. Questionnaire Survey

The Pet Attitude Scale (PAS) and Animal Attitude Scale (AAS) were used to collect demographic information, experience with keeping companion animals, and attitudes towards animals (see Appendix A for details).

#### 2.2.3. Experimental Procedure

The experimental procedure was programmed using Presentation 0.71 software with a block design. Participants first completed questionnaires and signed informed consent forms, and practiced outside the MRI room. The task required participants to make a button press decision when viewing the images. Once they understood the task, they entered the MRI room to begin the formal experiment. The practice session was identical to the formal experiment. The formal experiment included two sessions, consisting of a total of 24 blocks. Each block contained 8 images, with 6 blocks for each condition. There was an interval of 5–16 s (randomized) between blocks. The order of the blocks and the images within each block were presented randomly (see Appendix A for details).

### 2.3. Functional MRI Data Acquisition and Preprocessing

Data were collected using a Siemens 3T MRI scanner (Siemens, Erlangen, Germany). The scanner was equipped with a 64-channel head-neck coil. Task-based functional images were obtained using T2-weighted echo-planar imaging in an interleaved order (repetition time 2000 ms, echo time = 30 ms, field of view 224 × 224 mm^2^, flip angle = 90°, matrix size 224 × 224 mm, slice gap 0.3 mm, voxel size 2 × 2 × 2 mm^3^), acquiring 62 axial slices with a thickness of 2 mm covering the whole brain. Data were preprocessed using DPARSF 5.2, followed by spatial smoothing with a Gaussian filter of 6 mm full width at half maximum (FWHM) (see Appendix A for details).

### 2.4. Data Analysis

#### 2.4.1. Specific Activation Analysis for Processing Companion Animal Information

The preprocessed data were analyzed at the individual level using SPM12 (Wellcome Trust Center for Neuroimaging, London, UK, http://www.fil.ion.ucl.ac.uk/spm/software/spm12, accessed on 20 January 2024) with a general linear model (GLM). The design matrix included four task conditions as main predictors, covariates, and six head movement parameters. Group-level *t*-tests were subsequently conducted using DPARSF 5.2, with multiple comparison correction using Gaussian Random Field (GRF) at voxel-level *p* < 0.01 and cluster-level *p* < 0.05, two-tailed tests, and clusters larger than 30 voxels (see Appendix A for details). Since there were few male participants, we reanalyzed the data after excluding males and found that the results were essentially the same as those from the analysis of all data. Therefore, the results reported in this study include all data.

#### 2.4.2. Correlation Analysis Between Brain Activation and Questionnaire Data

DPARSF 5.2 was used to conduct correlation analyses between the brain regions specifically activated by companion animals and PAS and AAS scores, respectively. The positively correlated brain regions (*p* < 0.01) were saved as brain masks, and the blood oxygen level-dependent (BOLD) signal values were extracted and imported into SPSS (IBM SPSS Statistics 26.0) software to verify and quantify the correlation between neural activity and individual attitudes (see Appendix A for details).

#### 2.4.3. Exploratory Generalized PsychoPhysiological Interaction Analysis

To investigate whether connectivity changes between different brain regions were related to companion animal information processing, we conducted a Generalized PsychoPhysiological Interaction (gPPI) analysis [65]. The companion animal-specific activation results (right IPL, right MOG, left SFG, and left PCu) were selected as seed points for whole-brain functional connectivity analysis (*p* < 0.05 and cluster size > 50) (see Appendix A for details).

#### 2.4.4. Exploratory Dynamic Causal Modeling Analysis

To further investigate how companion animal information affects modulation between related brain regions and the intergroup differences between pet owners and non-pet owners. This study used Dynamic Causal Modeling (DCM) under the Parametric Empirical Bayes (PEB) framework to estimate effective connectivity within and between brain regions [66] (see Appendix A for details). In the DCM analysis of this study, we focused on: 1. Whether connectivity between certain brain regions was enhanced or weakened during companion animal processing. 2. Whether there were differences in connectivity between certain brain regions of pet owners and non-pet owners during companion animal processing. 3. Whether there were differences in intrinsic connectivity between brain regions of pet owners and non-pet owners in the absence of external stimuli or task conditions.

### 2.5. Auxiliary Tool—Usage of Generative Artificial Intelligence (GenAI)

During the preparation of this research manuscript, to enhance the accuracy and efficiency of literature analysis, the authors utilized two generative artificial intelligence (GenAI) tools: ChatGPT 4.0 and Doubao (Doubao, a GenAI tool developed by ByteDance; specific version not specified). The specific applications of these tools are as follows: 1. Assisting in understanding complex or ambiguous content in the literature: the interpretation function of these tools was utilized to preliminarily organize complex academic perspectives, which were subsequently verified and revised by the authors in light of their expertise in the field; 2. Assisting in organizing literature abstracts in the fields of human–companion animal neural interaction and mental health: following the preliminary extraction of core information from the literature, the authors further refined and synthesized the core theoretical perspectives contained therein. All AI-generated interpretive content and abstract collations were systematically reviewed and verified by the author team to ensure consistency with the original intent of the source literature and alignment of theoretical syntheses with academic logic. The authors bear full responsibility for the academic perspectives, data interpretation, and conclusion derivation presented in the final manuscript, and no unverified AI-generated content was included in the final manuscript.

## 3. Results

### 3.1. Demographic Variable Statistics

The demographic data of the participants in this study are shown in Table 1.

### 3.2. Specificity of Neural Mechanisms in Processing Companion Animal Information

During the processing of companion animal images, several brain regions exhibited specific neural activity levels (Table 2, Figure 2). Multiple comparisons were corrected using GRF, with significant clusters identified at *p* < 0.05, voxel level *p* < 0.01 (two-tailed test), and a cluster size threshold of >30. The regions identified were: the right Inferior Parietal Lobule (right IPL), the right Middle Occipital Gyrus (right MOG), the left Superior Frontal Gyrus (left SFG), and the left Precuneus (left PCu).

### 3.3. Correlation Between Brain Activation and Behavioral Data

Correlation analyses were performed between brain regions specifically activated by companion animal stimuli and scores from the Pet Attitude Scale (PAS) and the Animal Attitude Scale (AAS). A positive correlation was observed between right IPL activation in pet owners and PAS scores (r = 0.48, *p* = 0.013; Figure 3), but it became non-significant after GRF correction. No significant correlations were observed between AAS scores and activation in any brain regions specifically linked to companion animal stimuli.

### 3.4. Exploratory Generalized Psychophysiological Interaction Analysis

Brain regions with specific activation were used as seed points for whole-brain gPPI analysis. When the right MOG was used as a seed point, the functional connectivity between the right MOG and the left Cingulate Cortex (left CC), right ACC, left IPL, and left PCu showed specific coordinated activation during companion animal processing (Figure 4). No significant results were found when left SFG, right IPL, or left PCu were used as seed points. The results were subjected to Gaussian Random Field (GRF) correction, with *p* < 0.05.

### 3.5. Exploratory Dynamic Causal Modeling Analysis

To better understand the mechanisms underlying the functional connectivity observed in the gPPI results, we conducted additional exploratory analyses. Using the identified gPPI results (right MOG, left CC, right ACC, left IPL, left PCu) as regions of interest, we constructed a dynamic causal model within the PEB framework to determine if these regions exhibit specific effective connectivity during the processing of companion animal information.

DCM Result 1: When posterior probability (Pp) > 0.95, no significant changes in connectivity were observed between brain regions during companion animal processing, indicating that companion animal information did not significantly affect the self-modulation or mutual modulation of these regions.

DCM Result 2: When posterior probability (Pp) > 0.95, no significant differences were found in the connectivity of brain regions between pet owners and non-pet owners during companion animal processing.

DCM Result 3: Significant differences in intrinsic connectivity between brain regions were observed between pet owners and non-pet owners in the absence of external stimuli or task conditions (Figure 5). Specifically:

(1)The connectivity from the left IPL to the left PCu was significantly stronger in pet owners, with a difference of 0.008 compared to non-pet owners (Pp > 0.96) (Figure 5);(2)The connectivity from the right ACC to the right MOG was significantly weaker in pet owners, with a difference of −0.01 compared to non-pet owners (Pp > 0.97).

## 4. Discussion

This study used fMRI technology to investigate the specific activation of the brain during the processing of companion animal information. First, the study found that the right MOG, right IPL, left SFG, and left PCu exhibited specific neural activity distinct from other stimuli during the processing of companion animal stimuli. Secondly, gPPI analysis indicated that using the right MOG as a seed point, companion animal information processing had specific effects on the functional connectivity between the seed point and the left CC, right ACC, left IPL, and left PCu. Finally, DCM analysis showed that pet owners had significantly higher connectivity from the left IPL to the left PCu and significantly lower connectivity from the right ACC to the right MOG compared to non-pet owners.

Consistent with Hypothesis 1, the results of this study showed that companion animal information can elicit specific neural activity, involving brain regions related to attention, emotion, and attachment. Attachment theory suggests that individuals form strong social bonds because certain targets provide a sense of security and support [67]. Building on the foundational framework of attachment theory—which posits that strong social bonds form when targets offer security and support [67]—Keefer et al. [68] further elaborated that attachment relationships are not restricted to human interactions. They noted that non-human targets, such as companion animals, can also fulfill this security-providing role, aligning with earlier recognitions of attachment’s broader scope beyond interpersonal bonds. Moreover, existing studies [40,69,70] have confirmed that the relationship between humans and companion animals shares many similarities with the parent–child relationship—including both the advantages and mutual support characteristic of such bonds. Consistent with this, human attachment patterns are applicable to the formation and maintenance of attachment relationships between humans and companion animals.

Previous research has examined the neural basis of human social attachments, identifying three primary neural network systems—the reward-motivation system, embodied simulation/empathy system, and mentalization system—that jointly establish, maintain, and enhance our attachment relationships [70]. The authors suggest that human attachment involves complex higher-order processes, such as learning, memory, planning, and prediction. The reward-motivation system drives motivation and reward perception, enabling humans to encode reward expectations, evaluate emotional value, and adaptively adjust behavior [71,72,73]. The embodied simulation/empathy system facilitates the integration of interoception, mental representation, and emotional information in forming and maintaining attachment relationships, enhancing the understanding of others’ mental states and making interactions more immediate and emotionally engaging [70]. However, these mechanisms alone are insufficient for forming attachment relationships, as attachment requires not only basic reward, expectation, and interoception but also understanding others’ mental states and intentions. The mentalization system enables individuals to infer others’ beliefs and goals [74,75], fostering deeper emotional connections and interactions. This cognitive capacity allows individuals to consider others’ perspectives during social interactions, understand mutual needs, and ultimately form stable attachment relationships.

In this study, specific activations in the right IPL, right MOG, left SFG, and left PCu during the processing of companion animal information provide an intriguing perspective. These findings suggest that humans may engage neural mechanisms similar to those used in human social attachment when processing information about companion animals. Specifically, the IPL is associated with the embodied simulation/empathy system, while the PCu is linked to the mentalizing system. Notably, prefrontal-related brain regions are present across all three attachment neural network systems. Although the MOG is not part of the above-mentioned attachment neural networks, and studies on the occipital lobe’s role in attachment are relatively few, it may still contribute to the perception of attachment-related stimuli.

In previous studies, the occipital lobe was found to be active when processing complex visual stimuli, including face recognition [76], emotion perception [77,78], visual memory [79,80], and processing of dynamic scenes [81]. Although companion animals are different from conventional human emotional faces or general visual stimuli, they effectively activate the MOG, which is consistent with previous findings that the occipital lobe is involved in the preferential processing of animals or faces [82]. The perception and emotional response of humans to companion animal faces may have similarities in neural mechanisms to the processing of human facial expressions [26,83].

Attachment-related studies have found that mothers showed significantly greater activation in the occipital lobe when viewing photos of their own children compared to other children [84,85,86]. Furthermore, activation of the occipital cortex during tasks has been associated with attachment [44]. Additionally, children with reactive attachment disorder were found to have significantly smaller gray matter volume in the visual cortex compared to normal children, further indicating the role of the visual cortex in attachment [87]. Therefore, the specific activation of the right MOG during companion animal processing in this study may reflect a unique processing mechanism and emotional response in humans towards companion animals.

In parenting, the inferior parietal lobule, precuneus, prefrontal cortex, and occipital lobe are critical for the neural responses of mothers to infant cries and laughter, as well as processing attachment behaviors [88]. These brain regions are significantly activated when mothers observe their own infants compared to unfamiliar ones [84]. Beyond mother–infant interactions, activation in these four brain regions is also associated with prenatal oxytocin levels in expectant fathers and their beliefs regarding harmonious parenting [89]. The occipital lobe, parietal lobe, frontal cortex, and precuneus, activated during companion animal information processing, encompass neural networks responsible for visual processing, emotional regulation, inhibitory control [90], decision-making, and social interaction, all of which are essential for attachment responses [91].

In terms of lateralization, companion animals, as a positive stimulus, activated the left frontal lobe and left precuneus, consistent with previous findings on brain lateralization that the left hemisphere is associated with positive emotional processing [92]. Although the frontal lobe shows increased activity when regulating both positive and negative emotions, the left frontal lobe is more active than the right frontal lobe when processing positive emotions [93]. Furthermore, previous research has shown that left frontal lobe activation is related to attachment security in children [94], and in adults, left frontal lobe activation is associated with approach motivation in attachment contexts [95]. The left prefrontal–subcortical brain circuit is crucial for emotion and decision-making [96], and is also intimately linked to the formation of social attachment and interpersonal connections [97]. In this study, since there was no significant difference in emotional valence between companion animals and positive objects, significant specificity in neural activity was still observed in these brain regions during the processing of companion animal information after excluding other confounding factors. This suggests that specific neural mechanisms exist in the human brain for processing companion animal-related information, which may have evolved over time and are linked to human emotional attachment to companion animals.

Why does the aforementioned specific neural activity not occur when humans process information about other species or inanimate objects? Could this be related to “anthropomorphism”? Mota-Rojas [98] noted that the human brain subconsciously classifies companion animals as “human-like” entities, thereby triggering stronger emotional and cognitive engagement than that induced by other species (e.g., insects, reptiles) or inanimate objects. The core reason lies in companion animals’ ability to simultaneously satisfy two key prerequisites for anthropomorphism: morphological and behavioral similarity, and intimate emotional bonding. Inanimate objects lack facial expressions, movement patterns, or emotional feedback capabilities similar to those of humans. While some other species (e.g., farm animals, wild animals) possess certain biological traits, they are not categorized as “family members”—instead, they are mostly classified as having practical attributes or being environmental elements. This prevents the formation of emotional bonds of equal intensity, thereby making it difficult to activate targeted neural responses. Through long-term domestication, companion animals have developed physiological traits that form a “dedicated signaling system” adapted to human anthropomorphic perception—a key foundation not possessed by other species. Through shelter adoption experiments, Waller [99] found that dogs can accentuate their neotenic (i.e., paedomorphic) features via specific facial movements, and these expressions preferentially elicit human parental responses. From the perspective of facial muscle anatomical specificity, studies conducted by Burrows [100] and Sexton [101] further confirmed that dogs possess specialized facial muscles not found in other domestic animals (e.g., pigs, cattle) or wild canids (e.g., foxes). These muscles enable dogs to produce facial expressions analogous to human expressions of “grievance” and “appeal for help.” While wild canids (e.g., foxes) may occasionally possess similar muscles, these muscles are poorly developed and cannot be actively controlled. Other species (e.g., birds, reptiles) even lack such muscle structures entirely. This anatomical difference enables companion animals to continuously transmit to humans visual signals that are “susceptible to anthropomorphic interpretation,” thereby activating brain regions associated with vision, emotion, and attachment (e.g., MOG, IPL, PCu). In contrast, other species cannot establish effective signal transmission due to the absence of this anatomical basis—likely contributing to their inability to elicit the aforementioned specific neural activity. Evolutionary research by Kaminski [102] further explains this specificity at its root: the facial muscle anatomy and expressive capabilities of dogs are direct outcomes of “anthropomorphic preference-based selection” during human domestication. During long-term coexistence with dogs, humans prioritized the selection of individuals capable of producing “human-like expressions,” thereby gradually reinforcing the specialized traits of their facial muscles. In contrast, other species—including undomesticated wild animals and domestic animals raised for utilitarian purposes—have not undergone such selection and thus have not developed physiological and behavioral traits adapted to human anthropomorphic perception. This “evolutionary-level adaptability” ultimately results in only companion animals being able to trigger human anthropomorphic cognition, thereby activating specific neural activities associated with visual processing, emotion, and attachment. In contrast, other species or inanimate objects cannot elicit equivalent responses, as they lack the aforementioned adaptive foundations at the anatomical, behavioral, and evolutionary levels.

Previous research has found that neural activity may be modulated by the subjective experience of attraction or attachment to animals [44]. In this study, a positive correlation was found between right IPL neural activity in pet owners and PAS scores during the processing of companion animal information, which was consistent with Hypothesis 2, though it was not significant after correction. Therefore, whether neural activity in certain brain regions correlates with individuals’ attitudes towards companion animals requires further exploration.

In the seed-based exploratory whole-brain functional connectivity analysis, processing companion animal information specifically affected brain region functional connectivity. Compared to other conditions, processing companion animal information significantly enhanced functional connectivity between the right MOG (seed point) and the left CC, right ACC, left IPL, and left PCu. These findings align with the brain regions specifically activated by companion animals, including the occipital, frontal, parietal lobes, and precuneus. This suggests that processing companion animals engages a broad brain network, spanning from attention mechanisms to higher-order cognitive functions. These results are consistent with Hypothesis 3.

In the subsequent exploratory Dynamic Causal Modeling (DCM) analysis, with the right MOG, left CC, right ACC, left IPL, and left PCu as regions of interest, the effective connectivity between brain regions was not affected by the processing of companion animal information. Notably, in the initial stage of the DCM analysis, our primary focus was to investigate changes in effective connectivity between relevant brain regions during companion animal processing (DCM Result 1). However, the results indicated no significant changes in effective connectivity between brain regions during companion animal processing compared to other task conditions. Given the lack of significance in DCM Result 1, and previous research indicating an association between companion animal ownership and cognitive performance as well as neural activity [44,103], we decided to further explore the differences in companion animal information processing between pet owners and non-pet owners (DCM Result 2), as well as the differences in intrinsic connectivity between brain regions without external stimuli or task conditions (DCM Result 3). Further exploratory analysis revealed differences in intrinsic effective connectivity between pet owners and non-pet owners. Compared to non-pet owners, pet owners exhibited significantly increased intrinsic connectivity from the left IPL to the left PCu (Pp > 0.96) and significantly decreased intrinsic connectivity from the right ACC to the right MOG (Pp > 0.97).

The parietal lobe plays an important role in memory [104,105,106] and is also related to attention allocation [107,108], sensory information [109,110], and emotional responses [111,112]. Furthermore, IPL and PCu are both components of the Default Mode Network (DMN) [113], playing a role in the processing of social affiliation within the network [114], as well as information-driven and regulatory functions [115,116]. In previous research on children with Autism Spectrum Disorder (ASD), Wymbs et al. (2021) [117] found that reduced functional connectivity between the IPL and PCu was associated with deficits in practical and social skills. This may lead to difficulties for children in understanding and imitating actions, nonverbal communication, emotional affect, and participating in social interactions [118]—skills that are crucial for establishing and maintaining attachment relationships [70,119,120]. Long-term care and rearing of companion animals can frequently activate a neural circuit involving the sequence: “understanding pets’ intentions → retrieving interaction memories → strengthening attachment bonds”. According to the principle of neural plasticity [121], the functional connectivity between frequently activated brain regions gradually enhances over time. Therefore, the enhanced connectivity of this neural pathway in pet owners may contribute to both the efficient interpretation of signals emitted by pets (e.g., faster recognition of a pet’s intention to “go out”) and the deepening of attachment bonds with their pets. Therefore, the present study found that stronger connectivity from the left IPL to the left PCu in pet owners may be associated with the experience of long-term companion animal ownership.

The ACC and MOG are part of the fronto-occipital network, and although they are far apart in the brain, they are capable of complex functional coordination and information integration [122]. In previous studies, reduced functional connectivity between the ACC and occipital lobe was thought to be related to depression in Alzheimer’s disease (D-AD), potentially affecting people’s perception and response to emotional stimuli [123]. Decreased fronto-occipital network connectivity has also been associated with obsessive–compulsive disorder, affecting the efficiency and coordination of the brain during inhibitory control tasks [124]. However, in the pet-owner population, this lower connectivity may have adaptive advantages. Prior research indicates that neurons in the brain can adapt to experiences and training [121], with their neuroplasticity being affected by alterations in these experiences or the surrounding environment [125,126,127]. Previous studies suggest that prolonged experience can modify brain structure and function [128,129], influencing cognitive abilities like memory [130]. There may be compensatory or regulatory mechanisms between brain functional networks [131]. Long-term experience of pet ownership may have prompted the brain to optimize resource allocation, reducing the reliance of the ACC on visual processing or its regulation. Previous research suggests that this regulatory mechanism is related to attachment, prompting the brain to allocate resources optimally, thereby reducing psychological and physiological burdens on individuals and conserving brain metabolic resources [132]. Non-pet owners, due to their relative unfamiliarity with companion animal visual stimuli, may rely more on the anterior cingulate cortex (ACC) to regulate visual processing in the middle occipital gyrus (MOG) (e.g., more deliberate discrimination of “whether a companion animal is aggressive”). In contrast, pet owners—being highly familiar with companion animal visual stimuli—have brains that optimize resource allocation, reducing the ACC’s regulatory load; this change manifests as weakened ACC-MOG connectivity. Therefore, the lower connectivity observed in the pet owner population in the present study may be an adaptive regulation of neural activity in pet owners, which may be associated with their attachment to companion animals.

Several limitations should be taken into account when interpreting these results. First, the study had an imbalanced gender ratio. Prior research indicates gender disparities in attitudes and attachment to companion animals [133], and suggests that sex and gender could impact the functional connectivity patterns within the brain [134]. Future research should explore whether significant differences exist in specific neural activity between genders. Second, the study used static images of companion animals. In real-life interactions, companion animals provide dynamic visual, tactile, and auditory stimuli, which may jointly influence the neural mechanisms underlying emotion and cognition. Future studies should consider using more ecologically valid experimental conditions. Thirdly, the classification task (i.e., judging image categories) is characterized by relatively shallow task demands and may not fully activate the emotional and attachment-related neural systems. Finally, this study did not account for animal breed variability. For instance, humans typically exhibit a stronger preference or emotional bond with specific breeds, while so-called “dangerous breeds” may not elicit comparable levels of positive attraction. Furthermore, animal age also exerts a notable influence—compared with adult or aged animals, humans typically form a stronger emotional bond with juvenile companion animals.

## 5. Conclusions

In summary, our study indicates that the human brain exhibits specific neural activity when processing information about companion animals, involving brain regions associated with advanced cognition such as visual processing, emotion, and attachment, which are part of the human attachment network. The experience of raising companion animals may have a profound impact on the activity of the brain’s neural networks as an important factor. These findings provide a new perspective for understanding the neural mechanisms underlying human–companion animal interactions and offer scientific evidence for fields such as animal-assisted therapy.

## Figures and Tables

**Figure 1 animals-15-03162-f001:**
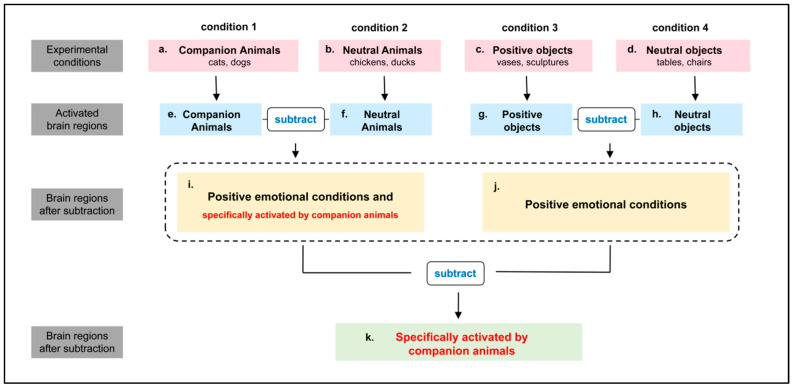
Analysis steps for fMRI results of companion animal-specific neural activity. (**a**–**d**) represent the four conditions in this study, (**a**) Companion animals (cats and dogs), (**b**) Neutral animals (chickens and ducks), (**c**) Positive objects (vases and sculptures), (**d**) Neutral objects (tables and chairs). On one hand, subtracting brain regions activated by neutral animals (**f**) from those activated by companion animals (**e**) yields brain regions activated by positive emotional stimuli and those specifically activated by companion animals (**i**). On the other hand, subtracting brain regions activated by neutral objects (**h**) from those activated by positive objects (**g**) yields brain regions activated by positive emotional stimuli (**j**). Finally, subtracting (**j**) from (**i**) yields brain regions specifically activated by companion animals (**k**).

**Figure 2 animals-15-03162-f002:**
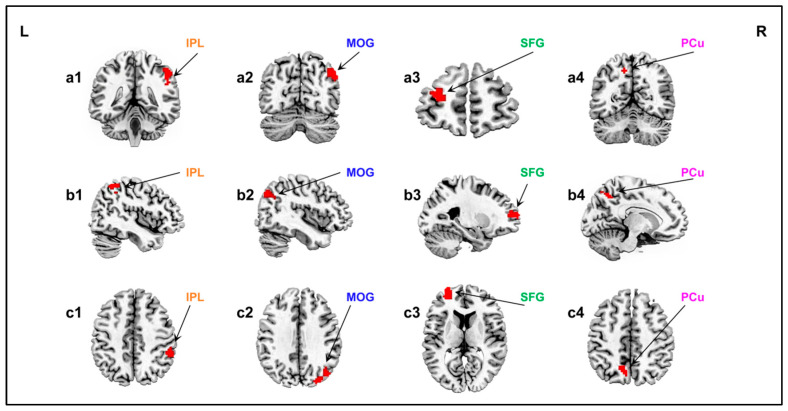
Specific Neural Activity in Processing Companion Animal Information. 1: Right IPL, peak value (42, −51, 54); 2: Right MOG, peak value (33, −84, 36); 3: Left SFG, peak value (−24, 60, 12); 4: Left PCu, peak value (−9, −63, 48). a: Coronal, b: Sagittal, c: Axial.

**Figure 3 animals-15-03162-f003:**
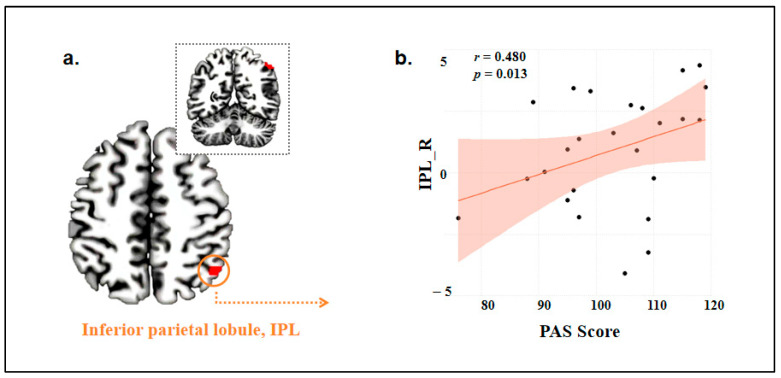
Correlation between right IPL specific activation and PAS scores in pet owners. (**a**) Brain region (right IPL) showing a positive correlation with PAS scores under companion animal processing in pet owners; (**b**) Correlation between right IPL blood oxygen level percentage and PAS scores, with PAS scores on the x-axis and blood oxygen signal percentage in the right IPL of pet owners on the y-axis.

**Figure 4 animals-15-03162-f004:**
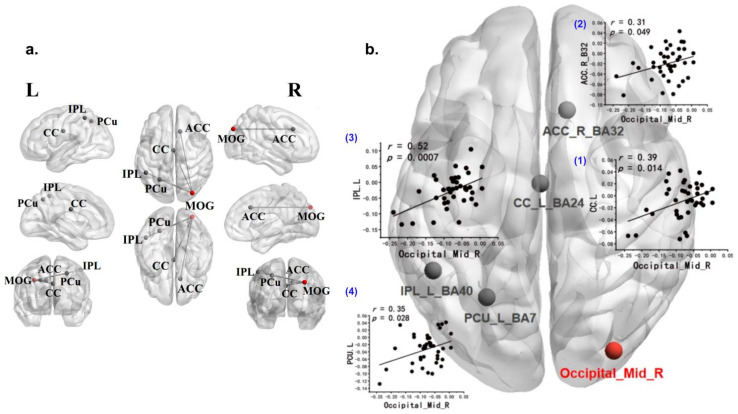
Generalized Psychophysiological Interaction Analysis (gPPI). (**a**) Brain regions exhibiting specific functional connectivity with the right middle occipital gyrus (MOG) during companion animal processing, with right MOG used as the seed point. (**b**) Functional connectivity between the seed point (right MOG, in red) and other brain regions: (1) left cingulate cortex (CC), r = 0.39, *p* = 0.014; (2) right anterior cingulate cortex (ACC), r = 0.31, *p* = 0.049; (3) left inferior parietal lobule (IPL), r = 0.52, *p* = 0.0007; (4) left precuneus (PCu), r = 0.35, *p* = 0.028.

**Figure 5 animals-15-03162-f005:**
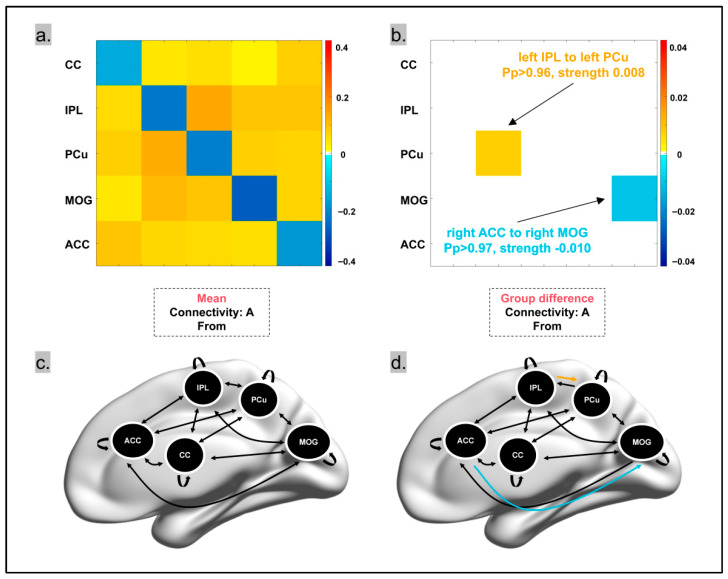
Mean effects and differences in intrinsic brain connectivity between pet owners and non−pet owners. (**a**) Mean matrix of intrinsic connectivity among regions of interest (ROIs). (**b**) Connectivity difference matrix between pet owners and non−pet owners. (**c**) Intrinsic connectivity among ROIs. (**d**) Differences in intrinsic connectivity between pet owners and non−pet owners, with arrows indicating the direction of signal transmission: black represents no difference, yellow indicates stronger connectivity in pet owners, and blue indicates weaker connectivity in pet owners.

**Table 1 animals-15-03162-t001:** Characteristics of Participants Included in This Study.

	N = 40
**Age (M ± S.D.)**	18–23 (19.6 ± 1.4)
**Gender**	
male	5
female	35
**Breeding experience**	
Pet Owner (PO)	26
Non-Pet Owner (NPO)	14
**Categories of companion animals**	
dog	16
cat	12
other mammals	4
reptile	3
insect	2
fish	1

**Table 2 animals-15-03162-t002:** Specific Activation in Processing Companion Animals.

Regions	Brodmann	Hemisphere	MNI Coordinates			Voxels	t
			x	y	z		
IPL	40	R	42	−51	54	104	3.90705
MOG	19	R	33	−84	36	84	4.19988
SFG	10	L	−24	60	12	68	3.70245
PCu	7	L	−9	−63	48	35	3.47122

## Data Availability

Data and material from all studies are freely available online on the Open Science Framework. https://osf.io/yadxn/files/osfstorage?view_only=597513d99a8045888d0439efbcb7c156 (accessed on 21 November 2023).

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
