# Peer review of "Specific Neural Mechanisms Underlying Humans’ Processing of Information Related to Companion Animals: A Comparison with Domestic Animals and Objects"

_animals, 2025, doi:10.3390/ani15213162_

Round 1

Reviewer 1 Report

Comments and Suggestions for Authors

Thanks for the opportunity to review this interesting article.

For the introduction I would recommend to briefly mention all relevant theories about human animal interactions - not only attachment theory.

Please give more information about the recruitment. 

Page 4: 

"All participants had normal or corrected-to-normal vision; no history of major illnesses, head trauma, claustrophobia, or any psychiatric, neu- rological, or cognitive disorders; no history of major surgery, no implanted metal or med- ical devices; and no fear of "cat, dog, chicken, duck" images." Please use several sentences to describe your inclusion and exclusion criteria or list them. 

Page 4: Please state the ethic code provided by your ethics committee. 

Regarding the chosen stimuli I have problems understanding the classification. Please give reasons why you chose chicken and ducks as neutral animals and also please justify how and why you classified the object stimuli. Please also refer to the "cultural characteristics" of the chosen stimuli - especially concerning the category positive items (maybe also as limitation).

There's no comma before and. 

Please state the version of SPSS you have been using. 

Page 7: Please give more information about the correction. 

300-302: In my opinion that is not recent research. We've known this for a long time. 

303-307: I would recommend to reformulate this paragraph, as it proven that the bond between handlers and their pets include similar advantages and disadvantages. At this point this paragraph sounds as it is something that is still questionable. 

339: too much space between "Frank & Sabatinelli"

431: space "interactions(Yang et al., 2024)"

433-437: I would not recommend to write "result" and change it to "might be associated with" as you have a very small sample size and it´s not possible to exclude the influence of other variables.

447: space "experiences and training(Akitake et al., 2023)"

450-451: space "can modify brain structure and function(Wang & Ke, 2023; Wang et al., 2023)"

460: "possibly resulting from their attachment to companion animals" see comment above.

Author Response

Comment1:  For the introduction I would recommend to briefly mention all relevant theories about human animal interactions - not only attachment theory.

Response1: We sincerely appreciate your valuable suggestions. In response to these comments, we have supplemented the Introduction section with additional relevant theories in the human-animal interaction field—including The Biophilia Hypothesis and Social Support Theory—moving beyond a sole focus on Attachment Theory. We also thank you for your professional guidance in enhancing the manuscript (see revised Lines 70–79).

Comment2: Please give more information about the recruitment. 

Response2: We have supplemented more detailed information on participant recruitment in the manuscript. (see revised Lines 156–172).

Comment3: "All participants had normal or corrected-to-normal vision; no history of major illnesses, head trauma, claustrophobia, or any psychiatric, neu- rological, or cognitive disorders; no history of major surgery, no implanted metal or med- ical devices; and no fear of "cat, dog, chicken, duck" images." Please use several sentences to describe your inclusion and exclusion criteria or list them. 

Response3: We have clearly organized the inclusion and exclusion criteria for participants in the manuscript, and have included more explicit criteria in the supplementary files. (see revised Lines 156–172).

Comment4: Page 4: Please state the ethic code provided by your ethics committee. 

Response4: We have supplemented the Ethics Approval No. (2023LS018) issued by the ethics committee to which this study belongs in the manuscript. (see revised Lines 171).

Comment5: Regarding the chosen stimuli I have problems understanding the classification. Please give reasons why you chose chicken and ducks as neutral animals and also please justify how and why you classified the object stimuli. Please also refer to the "cultural characteristics" of the chosen stimuli - especially concerning the category positive items (maybe also as limitation).

Response5: We have included a detailed explanation in the main manuscript and made more detailed and comprehensive supplementary materials available to provide additional clarification. Rationale for Classifying the Four Stimulus Types Employed in This Study For animal stimuli: Cats and dogs are classified as companion animals, as they align with the established definition of "companion animals" and form intimate emotional bonds with humans (e.g., companionship, emotional comfort). By contrast, chickens and ducks are primarily perceived as "economic animals" (i.e., providers of meat and eggs) or "environmentally associated animals" (e.g., common poultry in rural areas). Their interactions with humans are dominated by functional contact and lack intimate attachment, thus classifying them as neutral animals. For object stimuli: Vases and sculptures are characterized by core attributes of decorativeness and aesthetic value—they symbolize an elegant lifestyle, elicit aesthetic pleasure, and carry positive cultural connotations—and are therefore classified as positive objects. Tables and chairs, as utilitarian daily tools, serve core functions of fulfilling practical needs (e.g., supporting writing activities, enabling seating). Their emotional association with humans is weak, so they are classified as neutral objects.(see revised Lines 181-195).

Comment6: There's no comma before and. 

Response6: We have revised the manuscript.

Comment7: Please state the version of SPSS you have been using. 

Response7: We have added it in the text.

Comment8: Page 7: Please give more information about the correction. 

Response8: We have added it in the text. (see revised Lines 322-323).

Comment9: 300-302: In my opinion that is not recent research. We've known this for a long time. 

Response9: We have revised the expression to make the content more rigorous. (see revised Lines 371-385).

Comment10: 303-307: I would recommend to reformulate this paragraph, as it proven that the bond between handlers and their pets include similar advantages and disadvantages. At this point this paragraph sounds as it is something that is still questionable. 

Response10: We have adjusted the expressions in the manuscript to clearly present the confirmed conclusions and deleted the wording in the original expressions that might give rise to "remaining doubts". (see revised Lines 371-385).

Comment11: 339: too much space between "Frank & Sabatinelli"

Response11: We have removed them.

Comment12: 431: space "interactions(Yang et al., 2024)"

Response12: We have made the revision.

Comment13: 433-437: I would not recommend to write "result" and change it to "might be associated with" as you have a very small sample size and it´s not possible to exclude the influence of other variables.

Response13: Thank you for your attention to the rigor of the study and your professional suggestions! We have fully taken into account the limitations of this study, including the small sample size and the inability to completely rule out the interference of other variables, and have revised the absolute expressions related to "result" in the manuscript to "might be associated with"—a phrasing more in line with the actual situation of the study—to present the research findings more objectively and cautiously. (see revised Lines 557-559).

Comment14: 447: space "experiences and training(Akitake et al., 2023)"

Response14: We have made the revision.

Comment15: 450-451: space "can modify brain structure and function(Wang & Ke, 2023; Wang et al., 2023)"

Response15: We have made the revision.

Comment16: 460: "possibly resulting from their attachment to companion animals" see comment above.

Response16: We have made adjustments in the manuscript. (see revised Lines 585-588).

Reviewer 2 Report

Comments and Suggestions for Authors

General comments
The human-animal interaction has gained increasing relevance in current discussions about coexistence between humans and companion animals. It has been repeatedly suggested that such interactions offer general health benefits. Therefore, I consider this manuscript proposal to be innovative and suitable for this journal. However, a significant weakness lies in the manuscript’s lack of alignment with the journal’s submission requirements.
Response:

Another additional weakness is the lack of a clear relationship between the objective and the title, despite the presentation of various hypotheses. As a recommendation to the authors, it might be necessary to consider establishing a single hypothesis and several specific objectives to facilitate the understanding of the results.
Response:

Another important weakness is the improper citation format throughout the manuscript. Again, I encourage the authors to consult the author guidelines to avoid publication issues.
Response:

Particular comments

Line 2: I agree with the title; however, if the authors permit, I suggest modifying it to: "Comparison of Neural Mechanisms in Humans in Response to Exposure to Companion Animals, Domestic Animals, and Objects", as it may better reflect the intent of the study.
Response:

Line 9: According to the author guidelines, a simple summary must be included. Please consider adding it.
Response:

Line 11: Please indicate clearly that this is the study’s main objective.
Response:

Lines 12–13: I agree with the general description of the study; however, it would be helpful to mention the general characteristics of the participants (e.g., sex, average age, whether they were healthy individuals or had an emotional disorder). Additionally, it is important to specify whether they were randomly assigned to the different stimuli.
Response:

Line 15: Please specify the statistical analysis method used.
Response:

Lines 20–25: I understand that some results were purely descriptive and others were comparative. If the authors agree, I suggest including the exact p-values for the comparative results.
Response:

Lines 26–28: I do not agree with the conclusion as it diverges from the stated objective. If the authors agree, I suggest focusing the conclusion on the primary objective: to determine the neural processing involved in response to different stimuli.
Response:

Line 29: I agree with the suggested keywords; however, if the authors permit, I recommend including "human-animal bond" and "emotion" as additional keywords. This may increase the manuscript’s discoverability in different databases.
Response:

Line 39: If the authors accept my suggestion, I recommend including some examples of studies supporting the benefits of human-animal coexistence.
Response:

Lines 58–66: Please correct the font type used in this paragraph.
Response:

Line 68: Please correct the punctuation error.
Response:

Line 69: Include the original reference or correct the citation format, as it differs from the original citation.
Response:

Lines 95–114: In addition to my general comment, these paragraphs are confusing and the study's objective is unclear. This may be due to the presentation of multiple hypotheses. I suggest reducing the number of hypotheses and clearly stating the objective, aligning it with the suggested modifications.
Response:

Lines 127–128: Although the authors mention that the sample size was estimated, if possible, I suggest indicating the parameters considered for sample size calculation. If a formula was used, please describe it along with the corresponding reference.
Response:

Lines 131–134: It is unclear what the general inclusion and exclusion criteria for the participants were. If the authors agree, I suggest specifying the exact number of participants included and whether a specific sampling methodology was used (e.g., convenience sampling or full randomization). Please clarify.
Response:

Line 136: Please include the corresponding reference.
Response:

Line 137: Was the informed consent form included in the appendices? If so, please include the citation.
Response:

Line 139: Before this section, if the authors agree, I suggest adding a new section titled “Experimental Design” where the general characteristics of the study are described. For example, was this a prospective, blinded, randomized study?
Response:

Line 290: I understand these lines describe the most important findings, which are very interesting. However, if the authors agree, I suggest expanding on this idea: Why does this neural processing occur? Does it make sense that these brain regions are activated, based on current knowledge of their functions?
Response:

Line 332: I agree with the discussion at this point; however, if the authors permit, they could briefly address another important point: Why was this response not observed with other species or objects? Could this be related to anthropomorphism? I encourage the authors to explore this further, as it may represent the first neurobiological association of this phenomenon. I suggest consulting the following source to explore this topic further:
Mota Rojas. Anthropomorphism and Its Adverse Effects on the Distress and Welfare of Companion Animals. doi: https://doi.org/10.3390/ani11113263
Response:

Line 345: In line with the previous comment, this may be related to the anthropomorphism of human-animal relationships. If the authors permit, I also suggest discussing this further and consulting the following articles:

  • Waller et al. (2013). Paedomorphic Facial Expressions Give Dogs a Selective Advantage. doi: https://doi.org/10.1371/journal.pone.0082686
  • Burrows et al. Dog faces exhibit anatomical differences in comparison to other domestic animals. doi: https://doi.org/10.1002/ar.24507
  • Sexton et al. Raising an Eye at Facial Muscle Morphology in Canids. doi: https://doi.org/10.3390/biology13050290
  • Kaminski et al. Evolution of facial muscle anatomy in dogs. doi: https://doi.org/10.1073/pnas.1820653116
    Response:

Lines 461–470: I agree with the stated limitation. However, if the authors agree, I suggest including an additional limitation: the lack of consideration of the animals’ breed. For example, there is often greater affection or emotional connection toward certain breeds, while so-called "dangerous" breeds may not elicit the same level of attraction. Additionally, the animals’ age also plays a role, with stronger emotional bonds typically formed with younger animals compared to adults or older ones. Please consider this as another potential perspective for future studies.
Response:

Author Response

General comments
The human-animal interaction has gained increasing relevance in current discussions about coexistence between humans and companion animals. It has been repeatedly suggested that such interactions offer general health benefits. Therefore, I consider this manuscript proposal to be innovative and suitable for this journal. However, a significant weakness lies in the manuscript’s lack of alignment with the journal’s submission requirements.
Response: Thank you for your recognition of the manuscript's innovativeness and its suitability for the journal. We have adjusted and optimized the manuscript's title and core content to ensure they better meet the journal's submission requirements. Thank you for your valuable feedback and guidance !

Another additional weakness is the lack of a clear relationship between the objective and the title, despite the presentation of various hypotheses. As a recommendation to the authors, it might be necessary to consider establishing a single hypothesis and several specific objectives to facilitate the understanding of the results.
Response: We have optimized the problem of "unclear connection between the title and research objectives": on the one hand, we have reorganized the logical connection between the title and the core research direction to ensure that they are consistently focused; on the other hand, we have adjusted the description of the research design, revising the original multiple hypotheses into one core research hypothesis, and deriving two exploratory hypotheses based on this core hypothesis. Thank you for your important guidance in improving the logical coherence of the manuscript! (see revised Lines 120-133).

Another important weakness is the improper citation format throughout the manuscript. Again, I encourage the authors to consult the author guidelines to avoid publication issues.
Response: We have checked and revised the citation format throughout the manuscript in accordance with the journal's Author Guidelines, ensuring compliance with standards to avoid publication issues. Thank you for your guidance on the rigor of the manuscript's details!

Particular comments

Line 2: I agree with the title; however, if the authors permit, I suggest modifying it to: "Comparison of Neural Mechanisms in Humans in Response to Exposure to Companion Animals, Domestic Animals, and Objects", as it may better reflect the intent of the study.

Response: In light of your suggestion, we have decided to revise the title to”Specific Neural Mechanisms Underlying Humans’ Processing of Information Related to Companion Animals: A Comparison with Domestic Animals and Objects” — this revision highlights the subject and object while retaining the "specific neural mechanisms" element. (see revised Lines 2-4).

Line 9: According to the author guidelines, a simple summary must be included. Please consider adding it.
Response: We have added simple summary. (see revised Lines 10-22).

Line 11: Please indicate clearly that this is the study’s main objective.
Response: We have made the adjustments in the original text. (see revised Lines 26-28).

Lines 12–13: I agree with the general description of the study; however, it would be helpful to mention the general characteristics of the participants (e.g., sex, average age, whether they were healthy individuals or had an emotional disorder). Additionally, it is important to specify whether they were randomly assigned to the different stimuli.
Response: We have added the basic characteristics of participants in the original text. This study adopts a single-group design, and no participant grouping or intervention was conducted, thus it does not involve a randomized grouping process. All participants received the same stimulus tasks and data collection protocol. (see revised Lines 145-153)

Line 15: Please specify the statistical analysis method used.
Response: We have stated it in the abstract. (see revised Lines 31-39).

Lines 20–25: I understand that some results were purely descriptive and others were comparative. If the authors agree, I suggest including the exact p-values for the comparative results.
Response: We have added the p-values and Posterior probability (Pp) values in the manuscript. (see revised Lines 31-39).

Lines 26–28: I do not agree with the conclusion as it diverges from the stated objective. If the authors agree, I suggest focusing the conclusion on the primary objective: to determine the neural processing involved in response to different stimuli.
Response: We fully agree that the conclusion section should be more closely aligned with the study’s main objectives. We have made adjustments to focus on the core content of "exploring the neural processing mechanisms involved in responding to different stimuli," ensuring the conclusions are consistent with the research objectives. (see revised Lines 39-44).

Line 29: I agree with the suggested keywords; however, if the authors permit, I recommend including "human-animal bond" and "emotion" as additional keywords. This may increase the manuscript’s discoverability in different databases.
Response: We have added "human-animal bond" and "emotion" as keywords. (see revised Lines 49-50).

Line 39: If the authors accept my suggestion, I recommend including some examples of studies supporting the benefits of human-animal coexistence.
Response: We have added relevant research examples in the text. (see revised Lines 56-59).

Lines 58–66: Please correct the font type used in this paragraph.
Response: We have made the revisions.

Line 68: Please correct the punctuation error.
Response: We have made the revisions.

Line 69: Include the original reference or correct the citation format, as it differs from the original citation.
Response: We have made the revisions.

Lines 95–114: In addition to my general comment, these paragraphs are confusing and the study's objective is unclear. This may be due to the presentation of multiple hypotheses. I suggest reducing the number of hypotheses and clearly stating the objective, aligning it with the suggested modifications.
Response: We have optimized the expression of hypotheses in accordance with your comments, integrated the original multiple hypotheses into one core hypothesis (focusing on the specificity of neural mechanisms in humans' processing of companion animal information) and two extended exploratory hypotheses (centering on differences in brain region functional connectivity and the impact of pet ownership status). Meanwhile, we have clearly organized the research objectives to ensure logical consistency with the hypotheses and the title. (see revised Lines 121-134).

Lines 127–128: Although the authors mention that the sample size was estimated, if possible, I suggest indicating the parameters considered for sample size calculation. If a formula was used, please describe it along with the corresponding reference.
Response: We have incorporated the relevant references into the uploaded supplementary materials, updated the core parameters, formulas, and methodological bases for sample size calculation, and provided a corresponding explanation of these revisions in the main text. (see revised Lines 160-161).

Lines 131–134: It is unclear what the general inclusion and exclusion criteria for the participants were. If the authors agree, I suggest specifying the exact number of participants included and whether a specific sampling methodology was used (e.g., convenience sampling or full randomization). Please clarify.
Response: We have supplemented the specific inclusion and exclusion criteria for participants in the manuscript, clearly stated that this study included a total of 40 participants, adopted the convenience sampling method, and clearly explained the sampling process. (see revised Lines 146-154).

Line 136: Please include the corresponding reference.

Response: We have added the corresponding references in the text. Thank you for your guidance on the details of the manuscript!

Line 137: Was the informed consent form included in the appendices? If so, please include the citation.
Response: The informed consent form of this study has been provided as supplementary material for reference.

Line 139: Before this section, if the authors agree, I suggest adding a new section titled “Experimental Design” where the general characteristics of the study are described. For example, was this a prospective, blinded, randomized study?
Response: Although we did not set up a separate "Experimental Design" chapter, we have added a description of the overall characteristics of the study (including information related to the study type) in the "Methods" section to ensure the content is focused and logically coherent. (see revised Lines 146-154).

Line 290: I understand these lines describe the most important findings, which are very interesting. However, if the authors agree, I suggest expanding on this idea: Why does this neural processing occur? Does it make sense that these brain regions are activated, based on current knowledge of their functions?
Response: We have added content in the manuscript that, based on the principle of neural plasticity, elaborates on the differences: pet owners may strengthen relevant neural circuits due to long-term interaction, while non-pet owners may require more visual regulation due to unfamiliarity. This part demonstrates the causes of neural processing and the rationality of brain region activation. (see revised Lines 575-589).

Line 332: I agree with the discussion at this point; however, if the authors permit, they could briefly address another important point: Why was this response not observed with other species or objects? Could this be related to anthropomorphism? I encourage the authors to explore this further, as it may represent the first neurobiological association of this phenomenon. I suggest consulting the following source to explore this topic further:
Mota Rojas. Anthropomorphism and Its Adverse Effects on the Distress and Welfare of Companion Animals. doi: https://doi.org/10.3390/ani11113263
Response: We have supplemented the analysis in the Discussion section regarding the question of "why this neural response does not occur in other species or objects" and explored its association in combination with the anthropomorphism theory (Mota Rojas et al., 2021; https://doi.org/10.3390/ani11113263)—pointing out that humans' anthropomorphic tendencies towards companion animals may strengthen specific neural processing, while other species/objects lack such cognitive associations and thus do not exhibit similar responses. Thank you for your guidance on deepening the research perspective! (see revised Lines 460-505).

Line 345: In line with the previous comment, this may be related to the anthropomorphism of human-animal relationships. If the authors permit, I also suggest discussing this further and consulting the following articles:

  • Waller et al. (2013). Paedomorphic Facial Expressions Give Dogs a Selective Advantage. doi: https://doi.org/10.1371/journal.pone.0082686
  • Burrows et al. Dog faces exhibit anatomical differences in comparison to other domestic animals. doi: https://doi.org/10.1002/ar.24507
  • Sexton et al. Raising an Eye at Facial Muscle Morphology in Canids. doi: https://doi.org/10.3390/biology13050290
  • Kaminski et al. Evolution of facial muscle anatomy in dogs. doi: https://doi.org/10.1073/pnas.1820653116
    Response:We have further explored the potential impact of "anthropomorphism in human-animal relationships" on the research results in the Discussion section, and referred to the relevant literature you mentioned to enrich the analysis of this dimension. (see revised Lines 460-505).

Lines 461–470: I agree with the stated limitation. However, if the authors agree, I suggest including an additional limitation: the lack of consideration of the animals’ breed. For example, there is often greater affection or emotional connection toward certain breeds, while so-called "dangerous" breeds may not elicit the same level of attraction. Additionally, the animals’ age also plays a role, with stronger emotional bonds typically formed with younger animals compared to adults or older ones. Please consider this as another potential perspective for future studies.
Response: We have supplemented content regarding "the failure to include the breed and age factors of companion animals" in the Limitations section, explicitly mentioning differences in emotional bonding that may be caused by different breeds (such as so-called "high-risk" breeds and other breeds), as well as the characteristic that young animals are more likely to form strong emotional bonds compared to adult/elderly animals. We have also regarded this as an important perspective for future research expansion. (see revised Lines 590-606).

Reviewer 3 Report

Comments and Suggestions for Authors

This manuscrpt focuses on neural correlates of companion animal information with an fMRI. It is an interesting topic, though this study is limited in some areas.

  1. There were concerns over the generalizability of the modest-sized sample of 40 participants comprising a highly lopsided gender distribution (35 females and 5 males). The literature of emotional processing and attachment is filled with studies of gender differences and to overlook this imbalance would be an oversight.
  2. The use of static images of cats and dogs as stimuli is a poor proxy for real-life human-animal interaction. Attachment to a companion animal is a multisensory relationship that is experienced through touch, sound, and the interaction. It is a simplification to at least present this complex interaction as a process of passive viewing 2D images, and it tampers with the usefulness of the neural results.
  3. The classification task (judging image categories) is relatively shallow and may not fully engage the emotional/attachment systems the authors emphasize.
  4. The discussion fails to link the results in a meaningful manner by connecting activations on the occipital and parietal regions of the brain directly to the attachment systems. While some overlap exists, the evidence presented does not justify such strong claims.

Author Response

Comment 1: There were concerns over the generalizability of the modest-sized sample of 40 participants comprising a highly lopsided gender distribution (35 females and 5 males). The literature of emotional processing and attachment is filled with studies of gender differences and to overlook this imbalance would be an oversight.
Response 1: Thank you for your suggestion! We fully acknowledge the potential impact of the sample size (40 participants) and imbalanced gender distribution (35 females and 5 males) on the study's generalizability. This issue has been clearly mentioned in the Limitations section of the manuscript. We also recognize the importance of gender differences in emotional processing and attachment research—future studies will focus on optimizing the gender structure of the sample to enhance the generalizability of the results. Thank you for your reminder regarding the rigor of the research! (see revised Lines  591-595).

Comment 2: The use of static images of cats and dogs as stimuli is a poor proxy for real-life human-animal interaction. Attachment to a companion animal is a multisensory relationship that is experienced through touch, sound, and the interaction. It is a simplification to at least present this complex interaction as a process of passive viewing 2D images, and it tampers with the usefulness of the neural results.
Response 2: Thank you for your profound insights on the selection of stimulus materials! Static 2D images of cats and dogs cannot replace human-animal interaction in real scenarios—real attachment relationships rely on multi-sensory experiences such as touch and hearing, while passive viewing of static images indeed simplifies this complex process, which may affect the ecological validity of neural results. This issue has been clearly elaborated in the Limitations section of the manuscript. Future studies will consider adopting multi-sensory dynamic stimuli (such as interactive videos and real contact) to enhance the authenticity of the research. (see revised Lines  595-598).

Comment 3: The classification task (judging image categories) is relatively shallow and may not fully engage the emotional/attachment systems the authors emphasize.
Response 3: We fully agree that the categorization task of "judging image categories" lacks sufficient depth and may not fully activate the emotion/attachment system that the study focuses on. This issue has been supplemented and elaborated in the Limitations section of the manuscript. Future studies will consider designing tasks more in line with real attachment scenarios (such as emotional valence evaluation, interaction intention judgment, etc.) to better evoke the target neural mechanisms. (see revised Lines  598-600).

Comment 4: The discussion fails to link the results in a meaningful manner by connecting activations on the occipital and parietal regions of the brain directly to the attachment systems. While some overlap exists, the evidence presented does not justify such strong claims.
Response 4: We agree with the view that "the direct association between the activation of the occipital lobe, parietal lobe and the attachment system requires more sufficient evidence to support it", and adjustments have been made in the Discussion section regarding this issue—clearly stating that the current interpretation of the association between these brain regions and the attachment system still has limitations, and it is only a preliminary association based on existing studies; future research needs to further verify this with more multimodal data.

Round 2

Reviewer 3 Report

Comments and Suggestions for Authors

Effective responses to the proposed issues and relevant revisions have been made by the author, so the publication of this manuscript is recommended.